# A novel methodological approach to SaaS churn prediction using whale optimization algorithm

**Muhammed Kotan**[1], **Ömer Faruk Seymen**[2], **Levent Çallı**[1], **Sena Kasım**[1], **Burcu Çarklı Yavuz**[1], **Tijen Över Özçelik**[3]

**1** Department of Information Systems Engineering, Sakarya University, Sakarya, Turkey, **2** Department of Quantitative Methods, Sakarya University, Sakarya, Turkey, **3** Department of Industrial Engineering, Duzce University, Duzce, Turkey

\* ofseymen@sakarya.edu.tr

**Data availability statement:** The data underlying this study is part of a scientific

## Abstract

Customer churn is a critical concern in the Software as a Service (SaaS) sector, potentially impacting long-term growth within the cloud computing industry. The scarcity of research on customer churn models in SaaS, particularly regarding diverse feature selection methods and predictive algorithms, highlights a significant gap. Addressing this would enhance academic discourse and provide essential insights for managerial decision-making. This study introduces a novel approach to SaaS churn prediction using the Whale Optimization Algorithm (WOA) for feature selection. Results show that WOA-reduced datasets improve processing efficiency and outperform full-variable datasets in predictive performance. The study encompasses a range of prediction techniques with three distinct datasets evaluated derived from over 1,000 users of a multinational SaaS company: the WOA-reduced dataset, the full-variable dataset, and the chi-squared-derived dataset. These three datasets were examined with the most used in literature, k-nearest neighbor, Decision Trees, Naïve Bayes, Random Forests, and Neural Network techniques, and the performance metrics such as Area Under Curve, Accuracy, Precision, Recall, and F1 Score were used as classification success. The results demonstrate that the WOA-reduced dataset outperformed the full-variable and chi-squared-derived datasets regarding performance metrics.

## Introduction

Effectively managing customer churn poses a significant challenge within business operations, necessitating meticulous strategies to sustain and nurture relationships with potential customers [1]. In churn literature, a customer who left or is about to leave the company is regarded as a churned customer, whereas Huang et al. [2] define customer churn as the loss of valuable customers to competing firms. Bhattacharyya and Dash [3] state that poor core services, perceived excessive prices, inconvenience, competition, ease of service switching, core service failure, and negative word-of-mouth emerge as some of the heavily general factors on customers' churn decisions in the relevant literature. In this regard, the purpose of

research project conducted at the Sakarya University Institute of Natural Sciences and is owned by a private SaaS company. The name of the company will remain confidential. The data cannot be shared publicly due to its sensitive nature. Data are available for researchers who meet the criteria for access to confidential data. to request access, please contact bf@sakarya.edu.tr (SAKARYA UNIVERSITY Faculty of Computer and Information Sciences Scientific Publication Committee).

**Funding:** The author(s) received no specific funding for this work.

**Competing interests:** The authors have declared that no competing interests exist.

the Customer Relationship Management (CRM) department in companies is mainly aimed at retaining customers by understanding the needs and behaviors that may be churn in the future [4]. According to Gupta et al. [5], the cost of the retention actions of lost or potentially lost customers is lower than the cost of acquiring and developing a customer with the same profitability and loyalty to replace a lost customer, which is at least five or seven times more. It is predicted that a 5% decrease in customer churn rates will increase firm profitability by at least 20% [6].

Today, with the growing significance of data management, the popularity of data mining has surged. Consequently, artificial intelligence and machine learning solutions are now employed across various businesses. As a result, businesses are gradually utilizing data-driven methods to discern sector-specific factors affecting consumer churn. In this context, companies must pinpoint industry-specific factors that drive customer churn within their sector rather than focusing solely on general aspects. This approach ensures a strategic advantage. However, it is noteworthy that the literature predominantly encompasses academic and practical studies within B2C industries, such as telecommunication [2,7–10], retail [1,11,12], or banking [13–15], and there is a massive gap in customer churn studies carried out in the B2B field [16].

B2B considerations stand out in customer churn analysis due to churn manifestation in contractual and non-contractual scenarios. While industries primarily catering to B2C audiences – such as telecommunications, banking, insurance, and digital services – may witness the termination of service or product agreements unilaterally or bilaterally, the B2B landscape presents a distinct dynamic. Formal agreements between customers and companies within the B2B environment are absent, and these are characterized by non-contractual arrangements. Consequently, customers may discontinue their association with a company without prior notice. A notable B2B business model is software as a Service (SaaS), providing users with software accessibility via a cloud-based platform. This model encompasses essential features such as maintenance, data storage, security, and updates [17]. The global SaaS market is predicted to increase to $234,900 million in 2028, mainly due to the COVID-19 outbreak and the Russia-Ukraine War, up from $96,760 million in 2022 [18].

In the SaaS sector, new digital services and the proliferation of competitors have facilitated customers' ability to swiftly acquire the required services from various providers. Consequently, conducting churn customer analysis within the SaaS sector can yield favourable outcomes. Çallı and Kasım [19] highlight the limited number of studies on SaaS customer churn. They suggest that developing new models with diverse feature selection methods and predictive algorithms will significantly contribute to the literature and provide valuable insights for managers. In this regard, this study focuses on the enterprise resource planning (ERP) industry as a SaaS business model. A whale optimization algorithm-based model was used to detect potential churn customers by analyzing churn customers' past transactions and data.

Churn analysis often involves numerous independent variables (features), making interpretation complex, especially for decision-makers. This complexity necessitates high analytical proficiency and may require expert guidance or visual aids for meaningful insights. As an illustration, a range of studies in the domain of SaaS employ varying numbers of features. Sergue [20] utilizes 43 features, while Ge et al. [21] consider 21. Similarly, Amornvetchayakul and Phumchusri [22] use 23 features, and Çallı and Kasım [19] base their predictions on 16 features.

As a result, this research aims to showcase how optimization techniques can enhance precise predictions by utilizing a reduced set of features. This endeavour addresses a significant gap in the current literature. Specifically, the study compares predictive methodologies that

employ a wide range of features and optimization-driven approaches that utilize a more concise feature subset. Through this investigation, the goal is to demonstrate that optimization can deliver comparable, if not superior, levels of accuracy, which offers a more efficient and focused predictive approach suitable for real-world applications. Considering the gaps in the literature within the scope of the study, the research questions were formed as follows;

**RQ1** - What are the fundamental features of the SaaS in customer churn analysis?

**RQ2** - Can more efficient churn analysis models be created using optimization methods with fewer features?

The organization of this paper is as follows. After the Introduction we give a comprehensive Literature Review. The Methodology section outlines the approach used to apply WOA for feature selection, including data preparation and the experimental setup. Next, we discuss the Experimental Results, showcasing the performance of WOA and other feature selection techniques. The Conclusion summarizes the findings of the study. Finally, we discuss Limitations and Future Studies.

## Literature review

Our literature research consists of two parts. Customer Churn Studies, where intense research mainly focuses on B2C, are reviewed in the first part. Under the title of Customer Churn Studies in SaaS, a small number of studies, which we consider a significant gap in the literature, have been examined.

### Customer churn studies

This section covers the general literature on customer churn prediction and optimization algorithms. The customer churn studies carried out so far have been done mostly with descriptive analysis, predictive analysis, ensemble methods, and optimization algorithms. The descriptive analysis-based churn studies used Association Rules [10,23,24], Naive Bayes (NB) [8,25,26], and Clustering [27–29]. Predictive studies widely used Logistic Regression (LR) [30,31], Decision Trees (DT) [32–34], Support Vector Machines (SVM) [35,36], and Neural Networks (NN) [37–41] techniques. Moreover, deep learning techniques [42–45] were also used for churn prediction recently. Boosting [46–48], Random Forests (RF) [49–51] and Bagging [52–54] methods also were used widely for churn studies as ensemble methods.

Most churn studies considering the B2C field also use optimization methods to enhance prediction outcomes. For example, Faris [55] proposed a particle swarm optimization (PSO) model to increase the model's predictive power by adjusting the weights of the input features and optimizing the neural network structure together while handling imbalanced class distribution. The author claimed the PSO model was better at churn accuracy than state-of-art classifiers. Vijaya and Sivasankar [56] adjusted PSO for feature selection and simulated annealing (SA) for the churn prediction model. Then, the experimental results were compared with DT, LR, k-means clustering, SVM, and RF for accuracy, precision, and F-metrics. According to experimental results, the metaheuristic performance of the proposed model was more efficient and had better predictive levels. Al-Shourbaji et al. [57] proposed a model combines two metaheuristic algorithms, ant colony optimization (ACO) and reptile search optimization (RSO), for the feature selection. The authors then compared ACO-RSO model to standard ACO, standard RSO, PSO, multiverse optimizer (MVO), and Grey Wolf Optimizer (GWO). The ACO-RSO had higher accuracy with the minimum number of features. Manivannan et al. [58] developed a churn prediction model using GWO. The authors claimed that the accuracy and convergence of churn prediction of the GWO surpassed the results of ACO and PSO with less time. Pustokhina et al. [59] deployed a multi-objective rain optimization

algorithm to determine the optimal sampling rate of the synthetic minority over-sampling technique (SMOTE) and feature adjustment for optimal weighted extreme machine learning (OWEML). The model, which includes preprocessing, balancing, and classification of three different datasets, performed the best predictive performance among the compared models.

To our knowledge, previous research has yet to explore the utilization of optimization techniques in the relevant SaaS literature. In this context, our research process and findings are anticipated to address a fundamental gap in the existing literature by introducing an optimization approach for selecting optimal features, thereby contributing to improved prediction results.

## Customer churn studies in SaaS

The number of studies on churn in the SaaS sector is relatively low. That is because the cost of customer acquisition in the SaaS is relatively high compared to other sectors [21], thus limiting data sharing. Within the scope of the research, the literature review on customer churn analysis in SaaS is shown in Table 1. As mentioned before, the limited existing churn studies considering the SaaS field have been done using different techniques and algorithms. For example, Ge et al. [21] investigated a SaaS company dataset to predict whether a customer will churn in the next three months by analyzing twenty-one features of more than eight thousand customers. In the proposed model, Extreme Gradient Boosted (XgBoost) was applied to determine the defining features and to predict the churn customers and LR and RF were used for comparison. Amornvetchayakul and Phumchusri [22] conducted a churn prediction analysis for inventory management software with LR, SVM, RF, and DT. The study highlighted

**Table 1. Literature review on churn analysis in SaaS.**

| Author | Data Set | Predictive Algorithm | Best Scores | Feature Selection |
|--------|----------|----------------------|-------------|-------------------|
| [21] | · 8256 Samples <br> · 21 Features <br> · No Sector Information | · RF <br> · PCA <br> · XGBoost <br> · LR | · XGBoost: %75 | · No |
| [60] | · No Sample Information <br> · 5 Features <br> · Digital Marketing Campaign Management | · LSTM <br> · CNN <br> · SVM <br> · RF | · RF: %83 | · No |
| [22] | · 1718 Samples <br> · 23 Features <br> · Inventory Management | · DT <br> · LR <br> · SVM <br> · RF | · RF: %92 | · Chi-Square <br> · ANOVA |
| [20] | · 8869 Samples <br> · 43 Features <br> · Phone System and Call Center | · LR <br> · RF | · Models could not predict churn customers due to overfitting | · No |
| [19] | · 1951 Samples <br> · 10 Features <br> · ERP industry | · NB <br> · NN <br> · k-NN <br> · DT <br> · LR <br> · RF | · RF: %78 | · Chi-Square <br> · Gini Index <br> · Information Gain <br> · Gain Ratio |

*RF: Random Forest, PCA: Principal Component Analysis, XGBoost: Extreme Gradient Boosted, LR: Logistic regression, LSTM: Long Short-Term Memory Neural Network, CNN: Convolutional Neural Network, SVM: Support Vector Machine, DT: Decision Tree, NB: Naive Bayes, NN: Neural Network, k-NN: The k-Nearest Neighbours Algorithm*

that business metrics are the most important features in the data, and RF has better predictive performance.

In the study conducted by Sergue [20], considering the cloud-based phone system industry, no significant results were obtained from the predictive algorithms due to the imbalanced distributed real-life data that the large majority of customers in the dataset were not churned. The importance and balanced distribution of the dependent variable in the data set are clearly emphasized in this study.

Finally, a cloud-based ERP company was considered for customer churn analysis in the research conducted by Çallı and Kasım [19]. The RF algorithm gave the best result where different feature selection (FS) approaches were conducted. The findings show that the number of customers and the number of products are the most important features in customer churn analysis for the ERP company. The number of invoices and order variables follow the most important features with a relatively higher weight than the others.

## Methodology

This study utilized a real-world dataset to examine the seventeen distinct features of 1951 clients of an ERP firm in Germany and Türkiye. Notably, the foundational dataset shares similarities with previous research conducted by Çallı and Kasım [19]; however, the current research selectively analyzed 1100 observations following a rigorous data preprocessing process.

Seventeen features were used within the scope of the research. In Table 2, the customer identification information (CustID) and customer status (Status) fields, which are not included in the analysis but in the data, are also given.

The churn distribution was relatively balanced in the data set used within the research. As seen in Fig 1, 42.9% of the dataset's observations are churn, while 57.1% are not. The churn rates of the cleaned and balanced dataset obtained after the preprocessing steps are shown in Fig 1.

**Table 2. Data features and explanations.**

| Features | Explanations |
|---|---|
| CostID | Customer ID |
| Product | # of products |
| Customers | # of customers |
| Orders | # of orders |
| Offers | # of offers |
| Invoices | # of created invoices |
| Marketplace | # of marketplaces |
| SpecSoft | # of specialized software |
| CachRegister | # of cash registers |
| MailConnects | # of mail connections |
| Receipts | # of payment receipts |
| BaseReport | # of special reports |
| ProdOrder | # of production offers |
| Cargos | # of cargo usage |
| Payment | # of payment documents |
| UserCounts | # of users |
| Tickets | # of supports |
| Group | Origin of user company |
| Status | Whether churn or not (1 for churn, 0 for non-churn) |

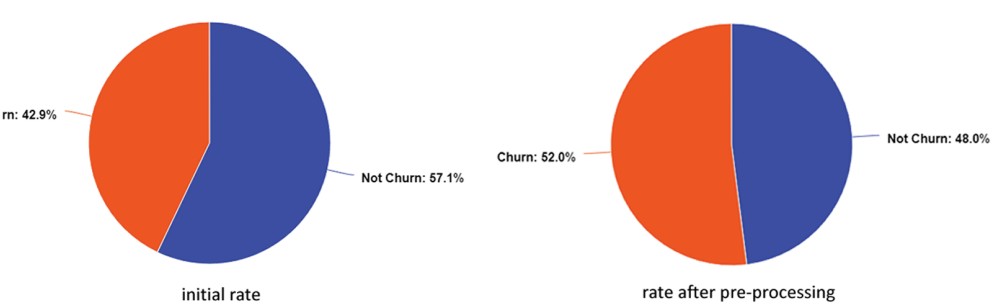

**Fig 1. Churn distribution of the SaaS dataset.**

The Whale Optimization Algorithm (WOA) [61] was applied in the FS stage to improve the accuracy of customer churn prediction utilizing k-NN, DT, NB, RF, and NN. Fig 2 illustrates the research process, mainly data processing, feature selection, prediction, and results.

## Feature selection

FS is one of the most commonly used and advantageous data preprocessing techniques. This technique involves identifying relevant features while eliminating irrelevant, redundant, or noisy ones. Removing irrelevant features found in real-world data through feature selection allows both storage and computational costs to be reduced without a significant loss of information or reduced learning efficiency [62].

Many FS methods have been proposed in the literature to achieve relevant features for classification and clustering purposes [63]. Metaheuristic optimization algorithms have become more popular due to their simplicity, ease of implementation, ability to bypass local optima, and applicability to various problems in different disciplines. As a result, many metaheuristic methods have been developed to deal with optimization problems in recent years. There is also an increasing use of advanced optimization techniques to search for the most suitable combinations of features [64].

In this research, we utilize the WOA [61] as a current technique for selecting potential features for churn analysis of the SaaS dataset. Recent studies have employed the whale algorithm for FS [65–69], including for churn analysis [70].

**Whale optimization algorithm (WOA).** Metaheuristic techniques can be employed to explore combinations of features within a dataset and select the best features. WOA mimics the hunting behavior of humpback whales and is inspired by the bubble-net hunting strategy

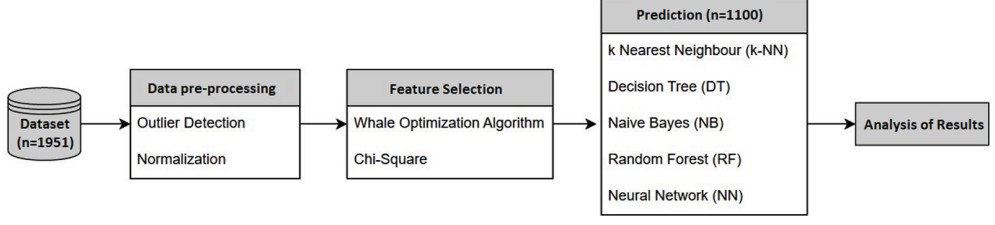

**Fig 2. Research process.**

[61]. The algorithm mathematically models the behaviors of humpback whales, such as encircling the prey, creating bubble nets, and searching for prey, to perform optimization tasks. The optimal solution is unknown in the search space, so the current best solution is considered optimal solution for each iteration. Following the detection of the best search agent, the remaining agents attempt to update their locations and mimic encircling the prey [71,72]. It is considered efficient due to its convergence rate and diversity preservation abilities during optimization [73]. The mathematical model of the three main steps of WOA is presented in Table 3, and the illustration of Bubble Net Attacking is shown in Fig 3.

• Encircling the prey: WOA assumes that the current best candidate solution represents the target prey or is near the optimal solution. Once the best search agent is identified, the other agents update their positions toward it. In this process, t represents the current iteration, A and C are coefficient vectors, Xgb denotes the position of the best solution found so far, and X is the position vector. The position of Xgb is updated in each iteration if a better solution is

**Table 3. Mathematical model.**

| Encircling The Prey | Bubble Net Attacking | | Searching The Prey |
|---|---|---|---|
| $A = 2a_1.r_1 - a_1$ | $X(t+1) = E.exp(bt).cos(2\pi l) + Xgb(t)$ | | $X(t+1) = Xr - A.D$ |
| $C = 2.r_2$ | $E = |Xgb(t) - X(t)|$ | | $D = |C.Xr - X|$ |
| $X(t+1) = Xgb(t) - A.D$ $D = |C.Xgb(t) - X(t)|$ | $X(t+1) = \begin{cases} Xgb(t) - A.D & p < 0.5 \\ E.exp(bt).cos(2\pi l) + Xgb(t) & p \geqslant 0.5 \end{cases}$ | | |

A and C; the coefficient vectors,
t; the iteration number,
r1 , r2 , l, p; random vectors in the interval [0,1],
a1; linearly decreased parameter;
E; the distance,
b; constant vector to determine the shape of the logarithmic spiral,
X; the position of the whale,
Xgb: the position of the best hunting agent.
Xr; the position of a randomly selected whale.

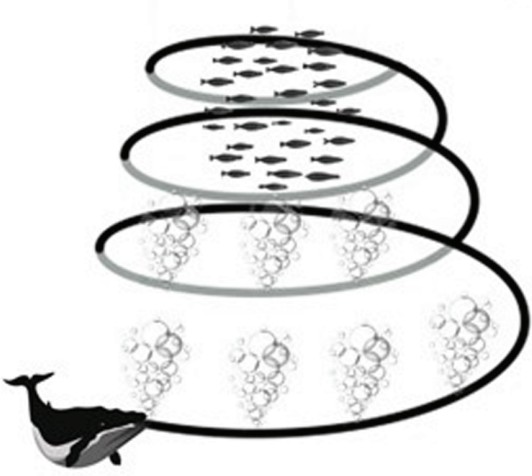

**Fig 3. Buble net attacking.**

found. For the calculation of A, $a_1$ decreases linearly from 2 to 0 over the course of the iterations, and r is a random vector in the range [0,1]. By adjusting the values of the A and C vectors, agents can explore different areas around the best solution. Search agents update their positions near the current best solution, simulating the encircling behavior of whales [61].

• Bubble-Net Attacking: To model bubble net behaviour, WOA uses a 50% probability to randomly choose between the shrinking encircling mechanism and the spiral model when updating the whale's position. The shrinking encircling mechanism is achieved by gradually decreasing the value of $a_1$. In the spiral updating position, the algorithm first calculates the distance between the current position of the search agent and the best solution. A spiral equation is then applied to mimic the helix-shaped movement of humpback whales. Here, E represents the distance of the i-th whale to the best solution found so far, b is a constant that defines the shape of the logarithmic spiral, l is a random number in the range [−1, 1] [61].

• Searching the prey: In the exploration phase, the variation of A is utilized to search for prey. Humpback whales search randomly based on the positions of other whales. To simulate this, random values for A are set to be greater than 1 or less than -1, enabling whales to move farther from a reference whale. In contrast to the exploitation phase, the position of a search agent is updated based on a randomly chosen whale from the population rather than the best solution found so far. Xr is a random position vector representing a whale chosen from the current population [61].

In WOA for feature selection, each feature subset consisting of features from the original set is represented as a position of a whale. The algorithm begins with a population of randomly generated feature subsets, and each solution is evaluated using the fitness function. The ideal solution is one that selects fewer features while maintaining higher classification accuracy [65]. WOA iterates through a series of steps in which the solutions update their positions, simulating the bubble-net hunting behavior and searching for prey. During each iteration, solutions adjust their positions based on either the shrinking encircling mechanism or the spiral model [65]. The shrinking encircling mechanism aims to balance exploration and exploitation using random vectors. The process continues until the stopping criteria are met.

In handling intricate optimization problems, Whale Optimization Algorithm (WOA) presents particular advantages compared to other optimization algorithms. First, WOA is more memory-efficient, as it only stores the best global solution at each iteration, reducing computational overhead [74]. Additionally, WOA exhibits better exploratory capabilities, enabling it to avoid local optima more effectively, which is crucial for problems with many local solutions [75]. Its adaptive mechanisms allow for accelerated exploitation based on iteration progress, leading to more accurate and refined solutions. Unlike many traditional algorithms, which rely on mechanisms such as crossover or mutation, WOA employs mathematical equations to update solutions, resulting in a more stable and efficient optimization process [74,76]. Furthermore, studies have shown that WOA offers faster convergence and higher solution quality than some other optimization algorithms [75].

**Chi-square.** Chi-Square is a statistical test used to compare the similarity of expected and actual model results [77] and to identify relationships between qualitative, categorical, or nominal variables in data [78]. The mathematical representation of the operational functionality of Chi-square is given as:

$$X_c{}^2 = \sum \frac{(O_i - E_i)^2}{E_i}$$

(1)

## Experimental results

In customer churn prediction, various advanced machine-learning approaches are utilized to model and forecast consumer behaviour. This section briefly overviews five widely recognized prediction methods: k-Nearest Neighbor, Neural Networks (NN), Random Forests (RF), Decision Trees (DT), and Naive Bayes (NB). k-NN algorithm has been used in churn studies due to its ability to create simple but powerful classifiers, built on a distance function that establishes how different or comparable two instances are [79]. To classify objects to the nearest output instance, for example, in a churn status, the k-NN classification method assumes that an unclassified target instance is similar to cases nearby in the feature space [80]. Even though k has a small positive value, an object can only be classified into one of its k closest classes if most neighbors agree [81]. In other words, the classification algorithm selects the test sample group from the k training samples closest neighbors to the test sample and assigns it to the class with the highest likelihood [82]. NN are well-liked machine learning methods that mimic the neurons' synaptic connections, which are human nervous system cells [83]. NN contains one or more intermediary additional layers called hidden layers, and the nodes embedded in these layers are called hidden nodes between the input and output layers [84]. The training performance of a NN is improved by adding more hidden nodes, but the generalization is frequently mediocre [85]. Breiman's [86] RF method starts with decision trees as the base classifier and then adds and blends randomness and bagging [87]. In each tree node, a subset of the features is chosen randomly, and the best split available within those features is chosen for that node [88]. Bagging is used to create the training set of data items for each tree in addition to randomness. RF is robust against overfitting and very user-friendly [15]. RF is robust to overfitting and can handle the missing values in the data used to build the predictive model [39]. DT is a popular modeling technique because of its strong performance and simplicity, which groups data from training data into nodes and branches in a hierarchical structure [89,90]. The decision tree assigns a chance for churn class in a classification tree to each leaf node and draws the target variable's values from a discrete domain [91]. A decision tree is constructed as a tree by adding child nodes along each branch [92]. The tree emerges when the leaf node ends or there is no information gain, and then the churn rules can be discovered by going through each branch [2]. NB classifier is aimed to calculate the probability values of each input belonging to a certain class in the Bayesian classification [2]. These are the probabilities of detecting churn and nonchurn in the context of the customer's churn [93]. The Bayes decision rule aims to place a new customer record in the class with the highest bayesian probability [94]. The NB classifier assumes that the existence (or absence) of any other feature has no impact on whether there is churn or not [95].

This section presents the classification and FS performance experiments using a WOA-based FS method on a new dataset containing the churn classification of SaaS customers. We obtained the results using the selected features from the WOA+kNN configuration (10 whales and 100 iterations were selected empirically). The algorithm was executed 50 times independently to ensure reliable analysis. Each solution has a fitness value that is used to evaluate potential solutions. The fitness function is assesses a solution's quality and improvement over earlier solutions. The evaluation classifier uses k-NN (k=5). We used 80% of the dataset as a training set and the remaining 20% as a test set. The fitness function is defined as:

$$Fitness = \alpha * (Err_{kNN}) + \beta * \frac{F_{Selected}}{F_{All}} \qquad (2)$$

where, $\alpha$ (0.99) and $\beta$ (0.01) are weighted factors that vary in the range of (0, 1). $Err_{kNN}$ is the ratio of misclassified instances by the k-NN classifier.

We present all dataset features, the top ten features from Chi-Square, and the features selected by the best three WOA+kNN implementations to five different classification methods, including k-NN, DT, NB, RF, and NN. We used 10-fold cross-validation to evaluate the performance of the selected methods. The codes were implemented using MATLAB 2022a on an i7-9700K CPU @ 3.60GHz PC with 32 GB RAM.

The results are summarized in Table 4. The table shows the performance results for all variables, chi-squared variables, and three different WOA-reduced data sets. The highest score in the corresponding row for each metric of each method is shown in bold. The data set names and the number of variables used in the customer churn analysis are shown in parentheses as WOA-1 (3), WOA-2 (5), and WOA-3 (10), respectively. According to Table 4, the WOA-based feature selection models have mostly higher scores on all performance measures than the all-features model or the chi-squared features model.

If we look at the metric results in the table; NB obtained the highest AUC score on the original data and NN on the chi-square. In other techniques, AUC scores were the highest in all WOA datasets.

The accuracy and precision metrics obtained the highest scores in WOA data sets in all techniques.

The recall metric obtained the highest score with RF and NN only in the dataset where all features were used, but the recall scores of the other techniques were obtained in the WOA datasets.

When the F1-scores, which better reflect the classification success in imbalanced data sets, are examined, it is seen that the data sets reduced with WOA perform better classification. F1-Score of the WOA-3 (10) data set yielded better in DT and RF techniques, whereas F1-Score

**Table 4. Performance scores of models.**

| Method | Performance Metric | All (17) | Chi-Square (10) | WOA-1 (3) | WOA-2 (5) | WOA-3 (10) |
|---|---|---|---|---|---|---|
| k-NN | AUC | 0.8324 | 0.8244 | 0.8297 | 0.8264 | **0.8353** |
|  | Acc | 0.7972 | **0.8000** | 0.7981 | 0.8054 | **0.8000** |
|  | Pre | 0.7510 | 0.7521 | 0.7543 | **0.7586** | 0.7550 |
|  | Rec | 0.9125 | **0.9178** | 0.9073 | **0.9178** | 0.9108 |
|  | F1 | 0.8239 | 0.8267 | 0.8238 | **0.8306** | 0.8256 |
| DT | AUC | 0.8342 | 0.8291 | 0.8460 | **0.8471** | 0.8387 |
|  | Acc | 0.8072 | 0.8072 | 0.8090 | 0.8136 | **0.8145** |
|  | Pre | 0.7593 | 0.7593 | 0.7585 | 0.7647 | **0.7651** |
|  | Rec | 0.9213 | 0.9213 | **0.9283** | 0.9265 | **0.9283** |
|  | F1 | 0.8325 | 0.8325 | 0.8349 | 0.8379 | **0.8388** |
| NB | AUC | **0.7825** | 0.7720 | 0.7755 | 0.7754 | 0.7783 |
|  | Acc | 0.6300 | 0.6272 | **0.6318** | 0.6309 | 0.6309 |
|  | Pre | 0.7894 | 0.7872 | **0.7950** | 0.7943 | 0.7943 |
|  | Rec | **0.3933** | 0.3881 | **0.3933** | 0.3916 | 0.3916 |
|  | F1 | 0.5250 | 0.5199 | **0.5263** | 0.5245 | 0.5245 |
| RF | AUC | 0.8410 | 0.8496 | 0.8514 | 0.8509 | **0.8581** |
|  | Acc | 0.7763 | 0.8009 | 0.7990 | **0.8072** | 0.8054 |
|  | Pre | 0.7063 | 0.7381 | 0.7503 | **0.7549** | 0.7445 |
|  | Rec | **0.9755** | 0.9562 | 0.9195 | 0.9318 | 0.9527 |
|  | F1 | 0.8193 | 0.8332 | 0.8263 | 0.8341 | **0.8358** |
| NN | AUC | 0.8360 | **0.8386** | 0.8373 | 0.8330 | 0.8286 |
|  | Acc | 0.7936 | 0.7954 | 0.7954 | **0.8036** | 0.7990 |
|  | Pre | 0.7379 | 0.7419 | 0.7419 | **0.7557** | 0.7468 |
|  | Rec | **0.9353** | 0.9300 | 0.9300 | 0.9195 | 0.9283 |
|  | F1 | 0.8249 | 0.8254 | 0.8254 | **0.8296** | 0.8277 |

of WOA-2 (5) data set was better in k-NN and NN techniques. F1-Score of WOA-1 (3) is only better in NB technique than in other datasets.

Another result from the table is that the AUC metric performances of the RF technique are better than other techniques when all data sets are considered.

Table 5 shows the confusion matrix of the RF technique (WOA-3(10 features)). When the results regarding the weights of the features are evaluated, it is seen that when the dataset is applied with the RF technique, the feature that contributes the most to the model is the number of customers (Table 6). Subsequently, the number of products and the number of marketplaces follow in order of decreasing importance. While number of the offers and number of the orders contributed to the model at the same rate with 0.062, number of the mail connections and number of the special reports contributed the least with 0.007 and 0.005, respectively.

The AUC-ROC graphs for five different data sets are shown in Fig 4, respectively. As can be seen in the graphs, the RF technique performs very well in all data sets according to the AUC scores as in all other performance metrics.

For further statistical validation, we employed the Friedman test to determine if there are significant differences in performance among the models. The Friedman test is one of the best non-parametric statistical tests to detect differences between multiple models across multiple datasets [96,97]. The test revealed below the significant level in the models' AUC performances (p= 0.001190), prompting us to conduct the Nemenyi post-hoc test to pinpoint the specific model pairs with significant differences. The results of the Nemenyi test were presented in a heatmap in Fig 5, with each cell displaying the p-value for a pairwise comparison between the two models. The color scale ranging from blue to red reflects the degree of significance, where blue cells denote lower p-values that highlight significant differences between models, and red cells indicate non-significant differences.

Obtaining approximate performances using fewer features revealed by examining the results shows that some features are more effective in churn decision-making for our data set. The WOA datasets had 3, 5, and 10 features, respectively. Notably that the 3 features in WOA-1 are also present in WOA-2 with 5 features and WOA-3 with 10 features. Therefore,

**Table 5. Random forest confusion matrix (WOA-3 (10)).**

|  | Predicted: Not Churn | Predicted: Churn | Total |
|---|---|---|---|
| **Actual: Not Churn** | 341(TN) | 187(FP) | 528 |
| **Actual: Churn** | 27(FN) | 545(TP) | 572 |
| **Total** | 368 | 732 | 1100 |

*TP: True Positive; TN: True Negative; FP: False Positive; FN: False Negative*

**Table 6. Random forest feature weights.**

| Feature | Weight |
|---|---|
| **Number of Customers** | 0.431 |
| **Number of Products** | 0.359 |
| **Number of Marketplaces** | 0.074 |
| **Number of Offers** | 0.062 |
| **Number of Orders** | 0.062 |
| **Number of Mail Connections** | 0.007 |
| **Number of Special Reports** | 0.005 |

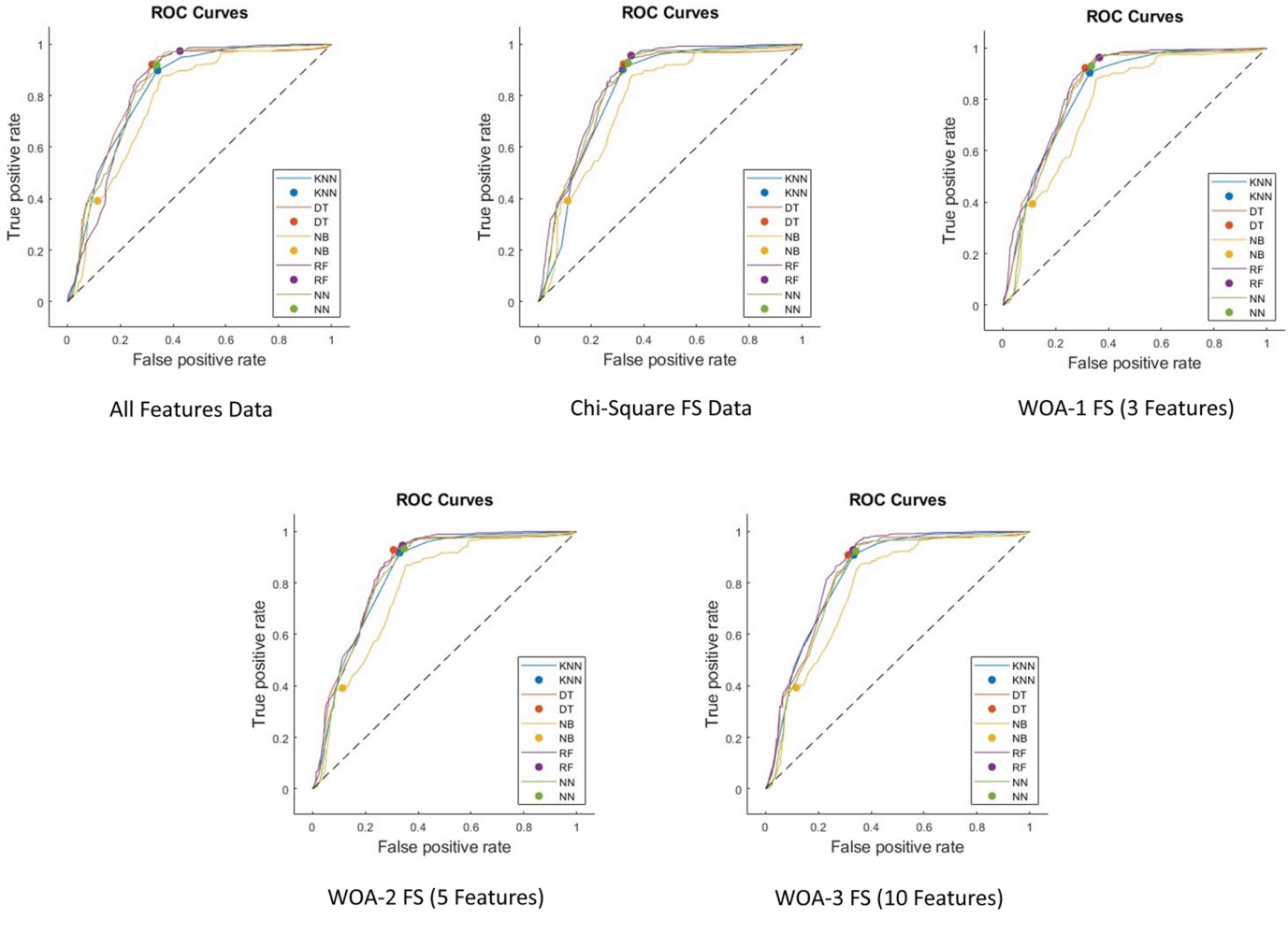

**Fig 4. The AUC-ROC graphs.**

the number of products, the number of customers, and the number of marketplaces are the most important variables in churn prediction in this SaaS dataset.

## Conclusion

Customer churn poses a significant challenge due to its impact on revenue and profitability. As such, companies prioritize retaining existing customers, as it is often more cost-effective than acquiring new ones. Customer churn is particularly critical for SaaS companies, as they usually operate in a non-contractual B2B model. Moreover, the high fixed costs of software development and maintenance mean that even a small rise in churn can substantially affect profitability.

The findings confirm that the study effectively identifies key features in SaaS customer churn analysis, using a real-world dataset and feature prioritization. Additionally, the findings show that optimization-based feature selection enhances both effectiveness and interpretability.

In this respect, our findings are consistent with the studies conducted in various sectors of SaaS, such as [21,22,60], emphasizing the significance of identifying distinct features for

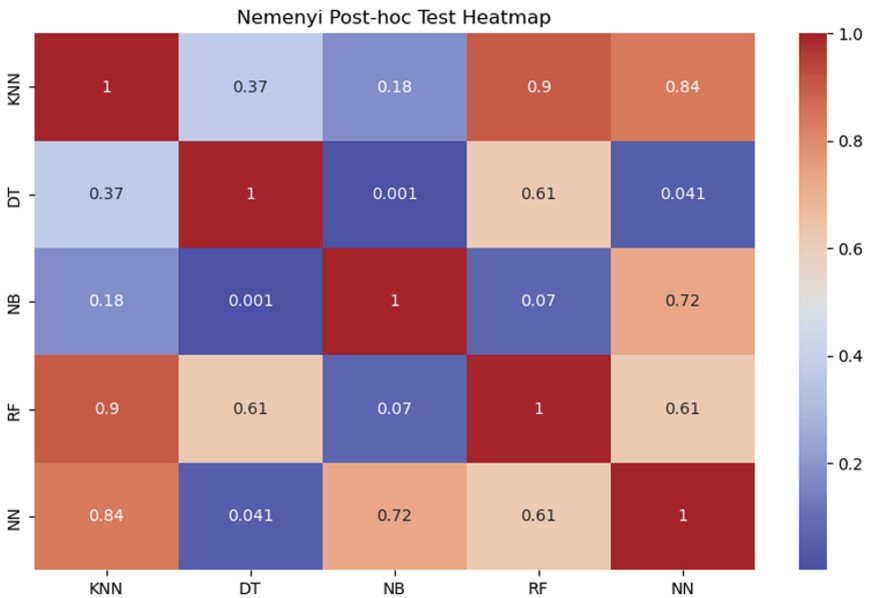

**Fig 5. Friedman-Nemenyi test for model comparison.**

each sector concerning customer churn analysis. In this sense, research findings have yielded a result that is believed to contribute to academic literature significantly. This study demonstrates that WOA-based feature selection can match or exceed the accuracy of models using all 17 features. In this regard, compared to the pioneering study of Çallı and Kasım [19], which utilized a similar dataset, our findings demonstrate that the effectiveness of data cleaning and optimization methods play a crucial role in shaping the results. Our study utilized optimization methods and identified the most significant features in the customer churn analysis model as the Number of Customers, Number of Products, and Number of Marketplaces. This result is believed to have contributed to the current academic literature by highlighting the significant differences that can arise from employing data preprocessing and optimization-based feature selection methods.

Regarding the practical contributions of the research findings, it is recommended that companies develop a customer churn analysis model tailored to the sector in which they operate, as customer behavior can vary even within sub-sectors. Furthermore, utilizing predictive models that require fewer features can provide management-level convenience and yield clearer interpretations of customer behavior. Based on the research findings, the most significant feature for a SaaS company operating in the ERP sector is the number of products users offer for sale through the system. Specifically, users with more products registered on the system are less likely to leave the SaaS service. As a practical implication, companies in this field may incentivize users to sell various products, encouraging continued use of their services.

Likewise, based on the research results, it can be concluded that SaaS users are more likely to continue using a company's services as the number of customers and sales across different marketplaces increase. Consequently, it is crucial for SaaS companies to strategically open new marketplaces and attract new customers through targeted promotional activities or by forming partnerships with other businesses.

## Limitations and future studies

This study encountered some limitations that can be addressed in future research. One of the limitations was the lack of consideration of transaction or web analytics data, which could have provided further insights into customer behavior. Incorporating these data types in future studies would be a significant academic contribution. Another limitation was that only the Whale Optimization Algorithm was used as the optimization method. Future studies could explore other optimization techniques, such as the Marine Predator Algorithm, Horse Herd Algorithm, Firefly Algorithm, or Grasshopper Algorithm, to compare their performance. These algorithms employ different search strategies and convergence speeds, which can offer advantages in feature selection processes by improving model efficiency and accuracy. Exploring multiple optimization techniques can help identify the most suitable approach for selecting the most relevant features, ultimately enhancing customer churn prediction.

Furthermore, the study focused solely on the ERP sector within the SaaS industry. Conducting similar research in different sub-sectors would provide additional insights and contribute to a more comprehensive understanding of customer churn in SaaS. Finally, different algorithms could be utilized for prediction purposes. Future research could compare the performance of different algorithms to determine the most effective method for predicting customer churn in SaaS.

## Author contributions

**Conceptualization:** Muhammed Kotan, Omer Faruk Seymen, Levent Calli, Tijen Over Ozcelik.

**Data curation:** Muhammed Kotan, Levent Calli, Sena Kasim.

**Formal analysis:** Muhammed Kotan, Omer Faruk Seymen, Tijen Over Ozcelik.

**Funding acquisition:** Sena Kasim.

**Investigation:** Sena Kasim.

**Methodology:** Muhammed Kotan, Omer Faruk Seymen, Levent Calli, Tijen Over Ozcelik.

**Project administration:** Omer Faruk Seymen, Tijen Over Ozcelik.

**Resources:** Sena Kasim.

**Software:** Muhammed Kotan, Burcu Carkli Yavuz.

**Supervision:** Omer Faruk Seymen, Levent Calli, Burcu Carkli Yavuz, Tijen Over Ozcelik.

**Validation:** Muhammed Kotan, Omer Faruk Seymen, Levent Calli, Burcu Carkli Yavuz, Tijen Over Ozcelik.

**Visualization:** Omer Faruk Seymen, Levent Calli, Burcu Carkli Yavuz.

**Writing – original draft:** Muhammed Kotan, Omer Faruk Seymen, Levent Calli, Burcu Carkli Yavuz.

**Writing – review & editing:** Muhammed Kotan, Levent Calli, Burcu Carkli Yavuz, Tijen Over Ozcelik.

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
