## [Decision Letter · Decision Letter 0]

4 Sep 2024

PONE-D-24-24312A Novel Methodological Approach to SaaS Churn Prediction Using Whale Optimization AlgorithmPLOS ONE

Dear Dr. Seymen,

Thank you for submitting your manuscript to PLOS ONE. After careful consideration, we feel that it has merit but does not fully meet PLOS ONE’s publication criteria as it currently stands. Therefore, we invite you to submit a revised version of the manuscript that addresses the points raised during the review process.

We look forward to receiving your revised manuscript.

Kind regards,

Ali Wagdy Mohamed

Academic Editor

PLOS ONE

Journal Requirements: When submitting your revision, we need you to address these additional requirements. 1. Please ensure that your manuscript meets PLOS ONE's style requirements, including those for file naming. The PLOS ONE style templates can be found at https://journals.plos.org/plosone/s/file?id=wjVg/PLOSOne_formatting_sample_main_body.pdf and https://journals.plos.org/plosone/s/file?id=ba62/PLOSOne_formatting_sample_title_authors_affiliations.pdf 2. Please note that PLOS ONE has specific guidelines on code sharing for submissions in which author-generated code underpins the findings in the manuscript. In these cases, all author-generated code must be made available without restrictions upon publication of the work. Please review our guidelines at https://journals.plos.org/plosone/s/materials-and-software-sharing#loc-sharing-code and ensure that your code is shared in a way that follows best practice and facilitates reproducibility and reuse. 3. In the online submission form, you indicated that your data is available only on request from a third party. Please note that your Data Availability Statement is currently missing contact details for the third party, such as an email address or a link to where data requests can be made. Please update your statement with the missing information. 4. Please review your reference list to ensure that it is complete and correct. If you have cited papers that have been retracted, please include the rationale for doing so in the manuscript text, or remove these references and replace them with relevant current references. Any changes to the reference list should be mentioned in the rebuttal letter that accompanies your revised manuscript. If you need to cite a retracted article, indicate the article’s retracted status in the References list and also include a citation and full reference for the retraction notice.

Reviewers' comments:

Reviewer's Responses to Questions

**Comments to the Author**

1. Is the manuscript technically sound, and do the data support the conclusions?

Reviewer #1: Yes

Reviewer #2: Yes

2. Has the statistical analysis been performed appropriately and rigorously? 

Reviewer #1: Yes

Reviewer #2: Yes

3. Have the authors made all data underlying the findings in their manuscript fully available?

Reviewer #1: Yes

Reviewer #2: Yes

4. Is the manuscript presented in an intelligible fashion and written in standard English?

Reviewer #1: Yes

Reviewer #2: Yes

5. Review Comments to the Author

Reviewer #1: Overall Assessment:

Your manuscript titled "A Novel Methodological Approach to SaaS Churn Prediction Using Whale Optimization Algorithm" presents a technically sound and innovative approach to customer churn prediction in the SaaS sector. The use of the Whale Optimization Algorithm (WOA) for feature selection and the comprehensive evaluation of multiple predictive models contribute significantly to the academic discourse on churn analysis. The study is well-structured, and the methodology is robust, making your findings valuable for both academic and practical applications.

1. Technical Soundness and Data Support: Your research is methodologically sound, with rigorous experimental design and execution. The comparative analysis of different datasets (WOA-reduced, full-variable, and chi-squared-derived) is thorough, and the use of multiple performance metrics provides a comprehensive evaluation of model effectiveness. The conclusions drawn are well-supported by the data, and the identification of significant features for churn prediction is both novel and practical.

2. Statistical Analysis: The statistical analysis in your manuscript is appropriately conducted. The use of 10-fold cross-validation adds rigor to your model evaluations, and the performance metrics employed are standard and relevant for this type of analysis. However, the inclusion of statistical significance tests (e.g., t-tests or ANOVA) to compare the performance of different models could further strengthen your analysis. While the current presentation of results is clear, adding these tests would provide additional confidence in the observed differences between models.

3. Data Availability: Your manuscript mentions that the data underlying the findings cannot be shared publicly due to privacy concerns but can be accessed by qualified researchers upon request. While this is understandable, it does not fully align with PLOS ONE’s data availability policy, which emphasizes the need for data to be fully available without restriction. To better comply with this policy, you might consider anonymizing the data or providing more detailed instructions on how researchers can request access to the data.

4. Presentation and Language: The manuscript is generally well-written and presented in standard English, making it accessible to a broad audience. However, there are a few minor grammatical and typographical errors that should be addressed during revision. Additionally, some sentences could be restructured to improve clarity and readability. Consistency in terminology, particularly regarding feature selection and dataset descriptions, would also enhance the manuscript's overall coherence.

5. Additional Comments:

Clarity in Methodological Explanation: While the methodological approach is generally clear, providing more detailed explanations of certain steps, such as the specific criteria used for feature selection with WOA, would further clarify your approach.

Ethical Considerations: There are no apparent ethical concerns regarding dual publication or research ethics based on the provided information. Your acknowledgment of privacy concerns related to data sharing is appropriate, though further clarification on how others can access the data is recommended.

Conclusion: Your manuscript makes a valuable contribution to the literature on SaaS churn prediction and feature selection methods. By addressing the minor language issues and considering the suggestions for data availability and statistical analysis, your manuscript would be well-positioned for publication. I encourage you to make these revisions to enhance the clarity, rigor, and compliance with journal policies.

Thank you for your contribution to the field.

Reviewer #2: The authors developed a method to SaaS Churn Prediction Using meta-heuristic: Whale Optimization Algorithm to identify critical variables rather than all variables affecting churn rates. The topic is interesting and the literature review is sufficient. The results are clearly presented. The paper could be considered for publication after minor revisions.

Some detailed comments:

Please give the organization of the paper.

Line 87 onwards: Please mention the paper author instead of just mentioning the reference numbers.

Line 153: Please mention the reference papers of WOA. Als mention why WOA is suitable for the problem being solved.

Table 3: Please write a paragraph explaining in details. The table is not sufficient for readers who are not familiar with metaheuristics

6. PLOS authors have the option to publish the peer review history of their article (what does this mean?). If published, this will include your full peer review and any attached files.

Reviewer #1: **Yes: **Mridula Dileepraj Kidiyur

Reviewer #2: **Yes: **Shuvodeep De

---

## [Author Response · Author response to Decision Letter 1]

6 Dec 2024

On behalf of the authors, I would like to thank you for your valuable comments and criticisms. We have our responses and additions to the issues raised in the “Responses to Reviewers” file that we uploaded through the revision system. You can also see the changes we have made on the article in our Track With changes file with highligts.

---

## [Decision Letter · Decision Letter 1]

12 Feb 2025

A Novel Methodological Approach to SaaS Churn Prediction Using Whale Optimization Algorithm

PONE-D-24-24312R1

Dear Dr. Seymen,

We’re pleased to inform you that your manuscript has been judged scientifically suitable for publication and will be formally accepted for publication once it meets all outstanding technical requirements.

Kind regards,

Pubudu Nuwanthika Jayasena

Academic Editor

PLOS ONE

Additional Editor Comments (optional):

I am pleased to inform you that your manuscript has been accepted for publication following the reviewer’s positive feedback. The reviewers have provided valuable insights and suggestions, which have greatly enhanced the quality of the manuscript. 

**Introduction:**Reduce repetition about the importance of churn analysis, and instead, emphasize the unique contribution of this study, such as applying WOA to the SaaS domain **Figures and Tables:**The inclusion of confusion matrices and AUC-ROC graphs is commendable. Ensure all visuals have concise and clear captions for better interpretation. **Language and Style**

Revise overly long or complex sentences for clarity.

**Future Work:**

Expand on potential future optimization techniques (e.g., Marine Predator Algorithm, Firefly Algorithm) and explain why they might outperform WOA.

Reviewers' comments:

Reviewer's Responses to Questions

**Comments to the Author**

1. If the authors have adequately addressed your comments raised in a previous round of review and you feel that this manuscript is now acceptable for publication, you may indicate that here to bypass the “Comments to the Author” section, enter your conflict of interest statement in the “Confidential to Editor” section, and submit your "Accept" recommendation.

Reviewer #1: All comments have been addressed

Reviewer #2: All comments have been addressed

2. Is the manuscript technically sound, and do the data support the conclusions?

Reviewer #1: Yes

Reviewer #2: Yes

3. Has the statistical analysis been performed appropriately and rigorously? 

Reviewer #1: Yes

Reviewer #2: Yes

4. Have the authors made all data underlying the findings in their manuscript fully available?

Reviewer #1: No

Reviewer #2: Yes

5. Is the manuscript presented in an intelligible fashion and written in standard English?

Reviewer #1: Yes

Reviewer #2: Yes

6. Review Comments to the Author

Reviewer #1: General Comments

The manuscript presents a well-executed and valuable study on SaaS churn prediction using the Whale Optimization Algorithm (WOA) for feature selection. It addresses a clear gap in the literature and demonstrates the advantages of optimization-based feature selection in improving model performance. The research is technically sound, supported by comprehensive experimentation, and appropriately framed within the context of existing studies. Below are comments and suggestions for further improvement.

Strengths

Novelty: The study contributes to the literature by applying WOA to SaaS churn prediction, offering insights into feature selection and optimization techniques.

Statistical Rigor: The use of cross-validation, AUC, F1-Score, and statistical tests (Friedman and Nemenyi) demonstrates a rigorous evaluation process.

Practical Implications: The findings have practical applications for SaaS providers in developing targeted retention strategies.

Transparent Methodology: The inclusion of details on dataset preprocessing, algorithm parameters, and experimental setup allows for reproducibility.

Major Comments

Data Availability:

The lack of publicly available data limits the ability of other researchers to replicate the study. Consider providing anonymized or synthetic datasets, if possible, to enhance transparency.

Alternatively, sharing detailed code or model pipelines could partially address this limitation.

Clarity in Results:

The Results section could benefit from more detailed explanations of the comparative performance of models, particularly how WOA-reduced datasets achieved better outcomes across metrics.

Include confidence intervals for the reported metrics to provide a clearer understanding of variability.

Overlong Descriptions:

Some sections, especially in the Methodology and Results, include overly detailed explanations of basic concepts (e.g., WOA mechanics). Condense these descriptions to maintain the reader's focus on the core contributions of the study.

Minor Comments

Abstract:

Simplify and focus on key findings. Avoid technical jargon where possible to increase accessibility.

Suggested Revision: "This study introduces a novel approach to SaaS churn prediction using the Whale Optimization Algorithm (WOA) for feature selection. Results show that WOA-reduced datasets improve processing efficiency and outperform full-variable datasets in predictive performance."

Introduction:

Reduce repetition about the importance of churn analysis and instead emphasize the unique contribution of this study (e.g., applying WOA to the SaaS domain).

Figures and Tables:

The inclusion of confusion matrices and AUC-ROC graphs is excellent. Ensure that all visuals are accompanied by concise, clear captions for easy interpretation.

Language and Style:

Revise overly long or complex sentences for clarity (e.g., in the Introduction and Conclusion). Example: "Reducing customer churn is a critical challenge for organizations as it significantly impacts revenue and profitability" can be simplified to "Customer churn poses a significant challenge due to its impact on revenue and profitability."

Future Work:

Expand on the discussion of future optimization techniques (e.g., Marine Predator Algorithm, Firefly Algorithm). Briefly explain why they might outperform WOA.

Overall Recommendation

The manuscript is a well-conducted study that offers valuable insights into SaaS churn prediction. With minor revisions to address clarity, conciseness, and data-sharing limitations, it will make a strong contribution to the field. Congratulations to the authors on this impactful work!

Reviewer #2: The authors have improved the manuscript based on my comments and suggestions. The paper could be accepted for publication.

7. PLOS authors have the option to publish the peer review history of their article (what does this mean?). If published, this will include your full peer review and any attached files.

Reviewer #1: No

Reviewer #2: **Yes: **Shuvodeep De

---

## [Editor Report · Acceptance letter]

PONE-D-24-24312R1

PLOS ONE

Dear Dr. Seymen,

I'm pleased to inform you that your manuscript has been deemed suitable for publication in PLOS ONE. Congratulations! Your manuscript is now being handed over to our production team.

Kind regards,

on behalf of

Dr. Pubudu Nuwanthika Jayasena

Academic Editor

PLOS ONE